# *Pleurotus highking* Mushroom Induces Apoptosis by Altering the Balance of Proapoptotic and Antiapoptotic Genes in Breast Cancer Cells and Inhibits Tumor Sphere Formation

**DOI:** 10.3390/medicina55110716

**Published:** 2019-10-28

**Authors:** Md. Anwarul Haque, Md. Anwar Ul Islam

**Affiliations:** 1Department of Experimental Pathology, Faculty of Medicine, University of Tsukuba, Tsukuba, Ibaraki 305-8575, Japan; a.haque5314@gmail.com; 2Graduate School of Comprehensive Human Sciences, University of Tsukuba, Tsukuba, Ibaraki 305-8575, Japan; 3Department of Pharmacy, University of Rajshahi, Rajshahi 6205, Bangladesh

**Keywords:** apoptosis, *Bax/Bcl-2* ratio, breast cancer, caspase 3/7 activity, flow cytometry, MCF-7 cells, tumor sphere, 3D culture medium

## Abstract

*Background and objectives*: Mushrooms that have medicinal properties are part of many traditional diets. The aim of the present study was to use the human breast cancer cell line MCF-7 to investigate the anticancer activity of *Pleurotus highking* mushroom purified extract fraction-III (PEF-III) and to elucidate the possible mechanism of that activity. *Materials and Methods*: The effects of PEF-III on cell proliferation and viability were evaluated by a colony formation assay and an MTT assay, respectively. Cell morphological changes, annexin-V phycoerythrin and propidium iodide (PI) staining, DNA fragmentation, and caspase 3/7 activity assays were performed to determine the induction of apoptosis by PEF-III. The genes responsible for regulation of apoptosis were analyzed by means of Western blot analysis. In vitro tumor sphere formation assay was performed using a 3D sphere culture system. *Results*: PEF-III significantly reduced the proliferation and viability of MCF-7 cells. Cell shrinkage and rounding, and annexin-V phycoerythrin and PI staining followed by flow cytometry indicated that the cell death was due to apoptosis. Additionally, a laddering DNA pattern and increased levels of caspase-3/7 enzyme also corroborated the notion of apoptosis-mediated cell death. This incidence was further confirmed by upregulation of proapoptotic genes (*p53* and its target gene, *Bax*) and downregulation of the expression of an antiapoptotic gene (*Bcl-2*). PEF-III also reduced the size and number of the tumor spheres in 3D culture conditions. *Conclusions*: The anticancer activity of PEF-III is due to induction of apoptosis by a shift in the balance of proapoptotic and antiapoptotic genes. Therefore, the findings of the present study may open a path to exploring potential drug candidates from the *P.*
*highking* mushroom for combating breast cancer.

## 1. Introduction

Cancer is one of the most devastating diseases, posing the threat of mortality to individuals worldwide despite promising advances in medical diagnosis and treatment [1]. To date, chemotherapy is one of the promising strategies to treat cancer. Numerous chemotherapeutic drugs isolated from natural sources, such as vincristine, vinblastine, bleomycin, paclitaxel, and camptothecin, are now used for cancer treatment. However, owing to the lack of specificity, low success rate, and risk of frequent recurrence associated with chemotherapy [2], new, safe, and effective anticancer drug candidates are urgently needed. Therefore, considerable attention has been focused on screening anticancer compounds from natural sources, including from medicinal plants [3].

As part of such a study, we focused on an edible and medicinal species of the oyster mushroom, the *Pleurotus highking*. From ancient times, oyster mushrooms have been considered throughout the world as a functional food and folk medicine [4]. They contain diverse biomolecules that are used for the treatment of various diseases including cancer [5]. Although a number of studies have already been conducted to explore the antibacterial and antifungal [6], anti-HIV [7], antitumoral [8], cytotoxic activity [9], antilipidemic [10], hyperglycemic and hypotensive [11], antiinflammatory [12], and immunomodulatory [13] properties of some species of oyster mushrooms, no reports have been published indicating the anticancer properties of the *P. highking* mushroom.

Chronic oxidative stress and inflammation are major risk factors for cancer development. Previous studies [9,12] reported that *P. highking* is a promising source of antioxidant and antiinflammatory agents. Hence, we hypothesized that, by supplying antioxidant and antiinflammatory metabolites, *P. highking* may reduce the risk of cancer. Here, we investigated the anticancer activity of *P*. *highking* purified extract fraction-III (PEF-III) on the human breast cancer cell line MCF-7 and explored the possible mechanism behind that.

## 2. Materials and Methods

### 2.1. Sample Collection

*P. highking* is commercially cultivated in Bangladesh. It was collected from the National Mushroom Development and Extension Centre, Savar, Dhaka-1340, Bangladesh. The mushroom was identified by an expert taxonomist at the Department of Botany, University of Dhaka, Bangladesh, and a specimen voucher of the sample under No. 20130123 was deposited in the National Herbarium, Bangladesh. Any dust was removed with clean water, and the mushrooms were dried in the shade for 12 days and stored in a closed container.

### 2.2. Extraction of Crude Mushroom Extract

Authenticated, shade-dried, and cleaned mushrooms were crushed into a fine powder by use of a grinding machine. About 250 g of the powder was soaked in 800 mL ethanol (95%) in an Erlenmeyer flask. The flask was then plugged with cotton and covered with aluminum foil for one week and stirred regularly. After incubation, the mixture was filtered, and the collected filtrate was then concentrated by use of a rotary evaporator under reduced pressure. A flask containing the concentrated extract was left in a vacuum dryer overnight.

### 2.3. Fractionation of Extract

The crude extract was fractionated by silica gel (60–230 mesh size) column chromatography using an increasing gradient of methanol in chloroform up to 100%. About 20 mL of every eluate was collected and combined with others that had similar compositions in thin layer chromatography. Five isolated fractions were dried using a rotary evaporator. A stock solution of the fraction was prepared by dissolving it in dimethyl sulfoxide (DMSO), which was further diluted with cell culture medium (DMEM; Dulbecco’s modified eagle media). Then, the test solutions were filtered through a 0.22 µm membrane filter and stored in −20 °C for further experiments.

### 2.4. Cell Line and Cell Culture

Cells from the breast cancer cell line MCF-7, obtained from the American type culture collection (ATCC), were cultured in DMEM and supplemented with 10% fetal bovine serum (FBS; Gibco, Waltham, MA, USA), 100 U/mL penicillin G, and 0.1 mg/mL streptomycin sulfate (Fujifilm Wako Pure Chemical Corporation, Osaka, Japan). The cells were incubated in a humidified atmosphere (5% CO_2_ at 37 °C). Before start of the experiments, the cells were checked for mycoplasma contamination.

### 2.5. Colony Formation Assay

About 400 MCF-7 cells were seeded in each well of a 6-well plate. After 24 h, the cells were treated at different concentrations (20 and 30 μg/mL) of PEF-III and incubated at 37 °C and 5% CO_2_ for 15 days. The cells were washed twice in PBS (Phosphate buffer saline), fixed in 4% formaldehyde, and stained with 0.01% crystal violet solution. Photographs of the fixed colonies were taken with a DSLR camera, and the colonies were counted by use of Image J software (National Institutes of Health, NIH). The following formula was used to calculate the colony formation rate: colony formation rate = (number of colonies after 15 days/number of cells seeded initially) × 100.

### 2.6. Cell Viability/Proliferation Assay

Cell viability was measured using the MTT (3-[4,5-dimethylthiazol-2-yl]-2,5-di-phenyltetrazolium bromide) assay. Briefly, 5 × 10^3^ cells/well were seeded in a 96-well plate and incubated at 37 °C and 5% CO_2_ with different concentrations (20 and 30 μg/mL) of PEF-III for 24 h. Then, MTT was added to the wells (100 μg/well), and the cells were incubated under the same conditions. An ELISA (enzyme-linked immunosorbent assay) microplate reader (BioTek Instruments, Winooski, VT, USA) was used to measure the absorbance at 595 mm. The results were expressed as the percentage of the control, which was considered to be 100%, and the plotted absorbance values were the means from three independent experiments.

### 2.7. Morphological Study Using an Inverted Light Microscope

The morphological study was performed according to a published report [14]. Briefly, 3 × 10^5^ cells were seeded in each well of 6-well plate and incubated for 24 h for attachment. Then, the medium was removed and fresh medium containing 20 and 30 μg/mL of PEF-III was added, and both the treated and the untreated cells were again incubated for 72 h. After incubation, morphological alteration of the cells was observed under an inverted light microscope (Olympus, Tokyo, Japan).

### 2.8. Annexin V-Phycoerythrin and PI Staining for Apoptosis

Next, to find out the causes of the anti-proliferative effect of the PEF-III, an apoptosis-indicating experiment, annexin V-phycoerythrin and propidium iodide (PI)- staining, was performed. First, the MCF-7 cells (1 × 10^6^) were plated in a 10-cm dish and incubated at 37 °C and 5% CO_2_ for 24 h. The cells were then treated with PEF-III and incubated for 2 days, after which they were washed in PBS twice, trypsinized, and resuspended in 100 μL of annexin-binding buffer (BD Company, Franklin Lakes, NJ, USA). After that, the resuspended cells were stained with annexin V-phycoerythrin (BD Company) and PI (BD Company) at a ratio of 1:10 and subjected to flow cytometry. FlowJo software (Tree Star, Ashland, OR, USA) was used for the apoptosis analysis.

### 2.9. Investigation of DNA Fragmentation

For confirmation of cell death through apoptosis, a DNA fragmentation assay was performed using the agarose gel electrophoresis method described previously [15]. In brief, about 1 × 10^6^ cells were seeded in each well and incubated for 24 h. The cells were then treated with different concentrations (20 and 30 μg/mL) of PEF-III and incubated again for 48 h. After the incubation, the cells were washed in PBS and resuspended again in PBS. Total DNA was isolated using a DNA extraction kit (Promega, Madison, WI, USA), analyzed by electrophoresis on 1% gel containing 0.1 μg/mL of ethidium bromide and visualized under a UV illuminator.

### 2.10. Caspase 3/7 Activity Assay

To investigate the mechanism of apoptosis, a caspase activity assay was performed according to the manufacturer’s protocol (Promega). First, 6 × 10^3^ cells/well were seeded in a 96-well clear-bottom white-walled plate containing 100 μL of DMEM, treated with different concentrations of PEF-III (20 and 30 μg/mL), and the cells were incubated for 24 h. Then, 100 μL of the assay kit was added and the cells were kept at room temperature for 2 h. Luminescence was measured using a fluorescence/multidetection microplate reader (BioTek Instruments).

### 2.11. SDS-PAGE and Western Blot Analysis

SDS-PAGE and Western blot analysis were performed according to the protocol described previously [16]. Briefly, cells were seeded in a 6-well plate, and after 24 h, treated with or without PEF-III (20 and 30 μg/mL) and incubated till 70–80% confluence. Lysates were prepared using lysis buffer (50 mM TrisCl, pH 7.8, 150 mM NaCl, 1% NP 40, 0.1% SDS, 1 mM phenylmethylsulfonyl fluoride). The lysates were then resolved on SDS-PAGE and transferred into polyvinylidene difluoride (PVDF) membranes. Next, the membranes were blocked in skim milk (4%) and Tris-buffered solution (50 mM Tris-HCl, pH 7.4, 150 mM NaCl, 0.1% Tween-20) for 1 h and incubated overnight at 4 °C with specific primary antibodies including anti-human Bcl-2 (Abcam, Tokyo, Japan), Bax (Cell Signaling Technology, Tokyo, Japan), p53 (Cell Signaling Technology), and β-actin (Cell Signaling Technology). After that, the membranes were incubated with horseradish peroxidase conjugated secondary antibodies for 50 min and then washed with Milli-Q (3–4 times), after which bound antibodies were detected by chemiluminescence reaction using Immuno Star Zeta (Wako) and EZ capture MG (ATTO Corporation, Tokyo, Japan) according to the manufacturers’ protocols.

### 2.12. In Vitro Tumor Sphere Formation Assay

Tumor sphere culture is a recently developed technique that is now widely used in drug screening for antitumor activity. Tumor sphere culture is a simple, inexpensive, and effective method for providing a physiological environment that closely resembles that of culture cells [17]. In vitro tumor sphere assay was performed according to the reported protocol [18]. Briefly, about 3 × 10^3^ cells were seeded in each well of a polyhema coated 6-well plate containing 3D sphere medium that contain either DMSO or PEF-III. The cells were incubated for 5 days and the tumor sphere size and number were calculated using an Olympus microscope.

### 2.13. Statistical Analysis

All the experiments were conducted in triplicate. The experimental data were expressed as means ± standard deviations (SDs) using Microsoft Excel software (iOS version 2011; Microsoft, Washington, DC, USA). When two values were compared (control vs. treatment), significance was assessed using the unpaired *t* test. Probability values lower than 0.05 were considered significant.

## 3. Results

### 3.1. PEF-III Decreases Proliferation of Breast Cancer Cells

To explore the anticancer effect of PEF-III on the breast cancer cell line MCF-7, proliferation of the cells was examined through colony formation and MTT assays. The PEF-III significantly reduced the colony-forming ability of the treated cells (21 and 42% at doses of 20 and 30 μg/mL, respectively) when compared with the untreated cells (Figure 1A,B). A similar trend was also observed in the MTT assay. When the cells were challenged with different concentrations of PEF-III, the number of viable cells was significantly decreased when compared with that of the untreated cells (Figure 1C). These results revealed that the PEF-III has strong anticancer activity against these treated breast cancer cells.

### 3.2. PEF-III Induced Apoptosis-Related Morphological Characteristics

The number of treated cells was significantly reduced at 72 h posttreatment when compared with the untreated cells. Cell detachment, cell rounding, cytoplasmic condensation, and cell shrinkage (Figure 2), which are the characteristics of apoptosis, were observed in the treated cells.

### 3.3. PEF-III Induces Cell Death through Apoptosis

To determine the mode of PEF-III mediated cell death, we performed a double-label staining annexin-V phycoerythrin and PI-binding experiment. As shown in Figure 3A, PEF-III induced apoptosis in a concentration-dependent manner, i.e., 76.5% and 81.7% apoptosis at doses of 20 and 30 μg/mL, respectively, as compared with the DMSO-treated cells. From these data, it was clear that PEF-III potentiates cell death by inducing apoptosis.

### 3.4. PEF-III Induces DNA Fragmentation and Increases Caspase 3/7 Activity

Further, to determine if PEF-III mediated cell death is due to apoptosis, a DNA fragmentation assay was performed. As shown in Figure 4A, a laddering pattern of genomic DNA was observed in the treated cells. These data indicate internucleosomal DNA degradation due to apoptosis. Additionally, to elucidate the molecular mechanism of apoptosis induced by PEF-III, we measured caspase 3/7 activity. As shown in Figure 4B, PEF-III increased the caspase 3/7 activity significantly when compared with the untreated cells. These data support the notion that PEF-III induced apoptosis through an intrinsic pathway.

### 3.5. PEF-III Alters the Expression of Proapoptotic and Antiapoptotic Genes

As apoptosis or programmed cell death is tightly regulated by various genes, we performed Western blot analysis to examine whether PEF-III affects the expression of proapoptotic genes such as p53, and Bax as well as the expression of antiapoptotic genes such as Bcl-2. The result revealed that in PEF-III treated MCF-7 cells, the expression levels of p53 and Bax were increased (Figure 5A) whilst the expression level of Bcl-2 was significantly decreased (Figure 5A and their corresponding densiometric analysis are shown in Figure 5B [*Bcl-2*], 5C [*Bax*], and 5D [*p53*]) as compared with the control. This result corroborates the notion that the mechanism by which PEF-III induces cell death through apoptosis is by altering the balance of proapoptotic and antiapoptotic genes.

### 3.6. PEF-III Inhibits Tumor Sphere Formation

In this study, we evaluated the effect of PEF-III on in vitro tumor sphere formation in a 3D culture system and found that PEF-III reduced the tumor sphere size (Figure 6A) and number (Figure 6B) significantly.

## 4. Discussion

Owing to the abundance of antioxidants and other medicinal constituents present in certain medicinal plants, intake of certain plant products is always inversely correlated with development of cancer [19]. In this study, we used *P. highking* extract fraction-III (PEF-III) to evaluate its anticancer activity against the breast cancer cell line MCF-7. We identified the mechanism of PEF-III mediated cell death and the genes responsible for that. Among five fractions, only fraction-III showed strong antiproliferative activity (data not shown) against breast cancer cells: its IC_50_ value was 24 μg/mL.

We found that PEF-III significantly reduced the colony numbers as compared with those of the untreated cells (Figure 1A,B). This was corroborated by a decrease in the number of viable cells during the MTT assay (Figure 1C). A similar result was reported by Andrej et al. [20], who used another species of oyster mushroom. Our data clearly show that the extract induced alteration of the growth kinetic in the tested cells.

To investigate the possible mechanism of cell death induced by PEF-III, we performed apoptosis-related experiments. First, we observed cell detachment from culture plates, cell shrinkage, and cell rounding (Figure 2) under an inverted light microscope in PEF-III treated cells. We also used double-label staining, annexin-V/PI followed by flow cytometry. Annexin-V usually binds with phosphatidylserine (PS). In healthy cells, PS is located on the inner side of the plasma membrane and annexin-V cannot bind with PS [21]. During the early apoptotic stage, PS translocates to the external membrane and annexin-V then specifically binds with PS and identifies the apoptotic cells. At the late stage, apoptotic cells and necrotic cells will stain positively owing to the passage of these dyes into the nucleus where they bind with DNA [22]. Our flow cytometric data (Figure 3A) revealed that PEF-III potentiates cell death by inducing apoptosis in a dose-dependent manner. Moreover, the DNA fragmentation (Figure 4A) and caspase 3/7 enzyme activation (Figure 4B) results led us to conclude that the cell death was due to apoptosis.

Cancer cells survive in a number of pathways. One common pathway is escape from apoptosis by downregulation of proapoptotic genes (*p53* and *Bax*) and hyperactivation of antiapoptotic genes (*Bcl-2* and *Bcl-xL*) [23]. Therefore, induction of apoptosis in cancer cells is considered a good strategy to treat cancer.

It is well known that a cell commits to apoptosis by altering the balance of proapoptotic and antiapoptotic genes [24]. As *p53* induces apoptosis through upregulating the expression of proapoptotic signals (such as Bax and BH-3 only members), *Bax* affects the integrity of the mitochondrial membrane through making pores in the membrane which leads to the release of cytochrome-c. BH-3 members directly bind with *Bax* to increase its activity and at the same time bind with *Bcl-2* (antiapoptotic signals) to inhibit its activity, ultimately leading to cell death [25,26].

In our study, we found that PEF-III upregulated the expression of *p53*, which is responsible for the upregulation of *Bax* and the downregulation of *Bcl-2* (Figure 5A) and hence increases the *Bax/Bcl-2* ratio, which promotes apoptosis and cell death. This finding suggests that a critical determinant of the overall propensity of cells underwent apoptosis as a result of the aforementioned treatment. From these data, it is clear that PEF-III induces apoptosis by shifting the balance between proapoptotic and antiapoptotic genes. In the tumor sphere formation assay, we observed that PEF-III treatment decreased the number of tumor spheres as compared with that of the untreated cells (Figure 6A,B).

## 5. Conclusions

Taken together, the findings of this study lead us to conclude that PEF-III possesses strong anticancer activity via induction of apoptosis by alteration of the balance of apoptosis-related genes. Therefore, further in-depth studies will be required to isolate the bioactive compound responsible for that activity. Additionally, we believe that this study will act as an eye-opener regarding the anticancer effect of *P. highking* and will serve as a base for future research to isolate a potent chemopreventive agent.

## Figures and Tables

**Figure 1 medicina-55-00716-f001:**
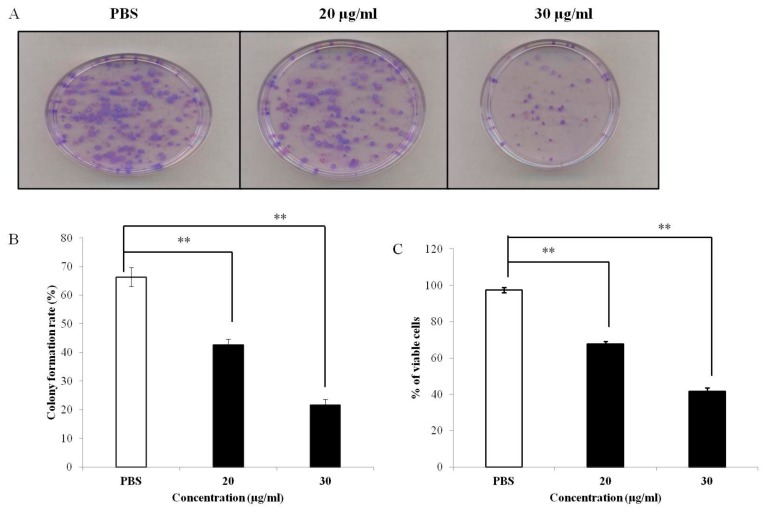
Inhibition of MCF-7 cell growth by *Pleurotus*
*highking* extract fraction-III (PEF-III). The cells were treated with different concentrations of PEF-III (20 and 30 μg/mL) and, incubated for 2 weeks, after which the colony was visualized by staining with crystal violate (**A**). The bar chart (**B**) depicts the quantification of the colony formation rate as compared with that of the control. After treatment with the indicated concentration of the extract, the cell viability was measured using the MTT assay (**C**). Values presented here are the means ± SDs of three independent experiments. Bars with asterisks indicate significant difference from the control at *P* ≤ 0.05 (*) or *P* ≤ 0.01 (**).

**Figure 2 medicina-55-00716-f002:**

Morphological changes of PEF-III challenged cells under an inverted light microscope. The cells treated with 20 and 30 μg/mL of PEF-III exhibited morphological changes, and apoptosis related characteristics such as cell shrinkage and rounding and detachment from the culture dishes were observed (100× magnifications).

**Figure 3 medicina-55-00716-f003:**
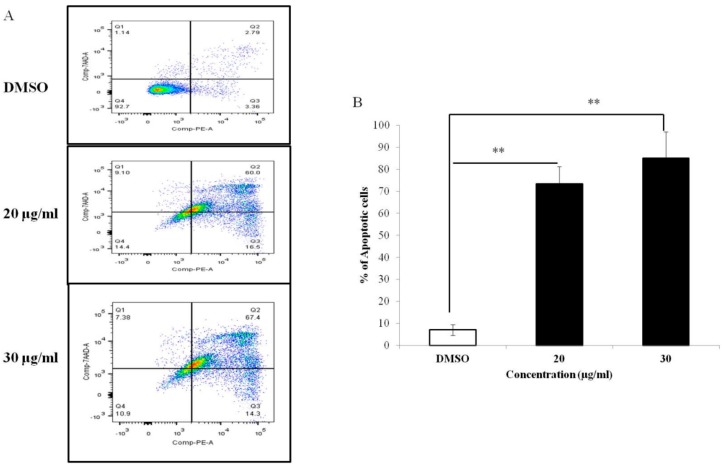
Apoptosis-inducing effect of PEF-III in the breast cancer cell line MCF-7. The cells were treated with different concentrations of the purified extract (20 and 30 μg/mL) and, stained with annexin V-PE and propidium, and apoptosis was measured by flow cytometry (**A**). The bar chart (**B**) shows the percentage of apoptotic induction by the extract as compared with that by the control. Values presented here are the means ± SDs of three independent experiments. The bars with asterisks indicate significant difference from the control at *P* ≤ 0.05 (*) or *P* ≤ 0.01 (**).

**Figure 4 medicina-55-00716-f004:**
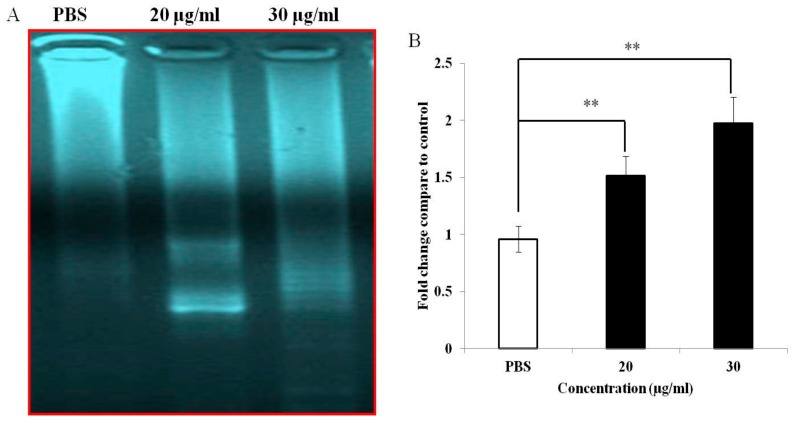
Effects of PEF-III on DNA fragmentation and caspase 3/7 activity in MCF-7 cells. Treated (20 and 30 μg/mL of PEF-III) and untreated cells were collected. The total genomic DNA of the cells was then run in 1% agarose gel containing 0.1 μg/mL ethidium bromide and visualized under a UV illuminator (**A**). Both the treated and the untreated cells were stained with aminoluciferin labeled substrate caspase, the lysates were incubated with caspase 3/7, and the luminescence (caspase 3/7 activity) was measured (**B**). Data are expressed as fold changes in comparison with the untreated (control) cells. Data for caspase 3/7 activity are presented as the means ± SDs of three independent experiments. The bars with asterisks indicate significant difference from the control at *P* ≤ 0.05 (*) or *P* ≤ 0.01 (**).

**Figure 5 medicina-55-00716-f005:**
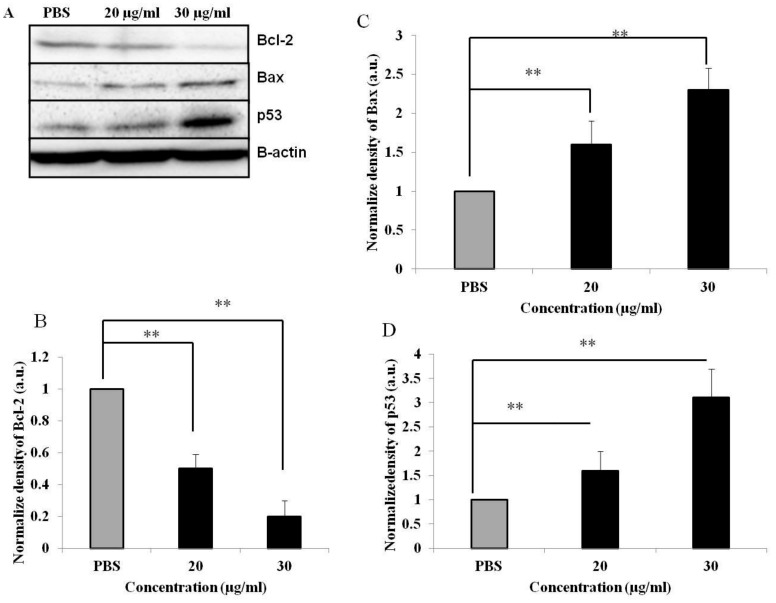
Western blot analysis of the expression of the proapoptotic and antiapoptotic genes. The expressions of the *Bcl-2*, *Bax*, and *p53* genes were investigated by means of Western blot analysis in MCF-7 untreated and treated (20 and 30 μg/mL of PEF-III) cells (**A**). The β-actin expression confirmed the loading control and was also used to normalize the expression. The quantitative bars (**B**–**D**) show the respective normalized expressions of the *Bcl-2*, *Bax*, and *p53* genes, which were measured by densitometry with Image-J software. Data are presented as the means ± SDs of three independent experiments. The bars with asterisks indicate significant difference from the control at *P* ≤ 0.05 (*) or *P* ≤ 0.01 (**).

**Figure 6 medicina-55-00716-f006:**
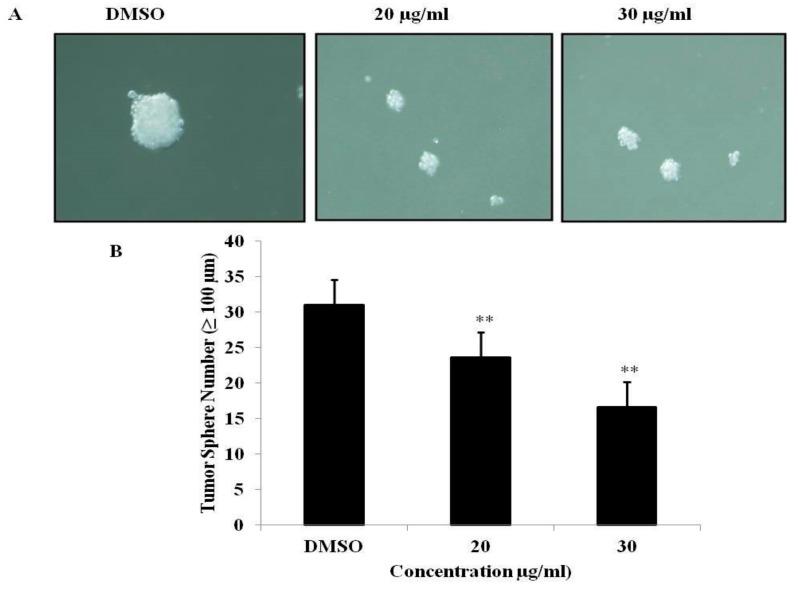
Effect of PEF-III on tumor sphere formation. Cells were cultured in a 3D sphere-forming medium with or without PEF-III and after 5 days’ incubation; the size (**A**) and number (**B**) of the tumor spheres were measured. Data are presented as the means ± SDs of three independent experiments. The bars with asterisks indicate significant difference from the control at *P* ≤ 0.05 (*) or *P* ≤ 0.01 (**).

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
