# Peer review of "Pleurotus highking Mushroom Induces Apoptosis by Altering the Balance of Proapoptotic and Antiapoptotic Genes in Breast Cancer Cells and Inhibits Tumor Sphere Formation"

_medicina, 2019, doi:10.3390/medicina55110716_

Round 1
Reviewer 1 Report
The work "Pleurotus highking Mushroom Induces Apoptosis by Altering the Balance of Proapoptotic and Antiapoptotic Genes in Breast Cancer Cells and Inhibits Tumor Sphere Formation" presents very valuable research on anti-cancer activities of Pleurotus highkingand perfectly fits into current research trends regarding the global problem of cancer. I recommend it for publication.
Author Response
Reviewer 1
Open Review
(x) I would not like to sign my review report
( ) I would like to sign my review report
English language and style
( ) Extensive editing of English language and style required
(x) Moderate English changes required
( ) English language and style are fine/minor spell check required
( ) I don't feel qualified to judge about the English language and style
|
Yes |
Can be improved |
Must be improved |
Not applicable |
|
|
Does the introduction provide sufficient background and include all relevant references? |
(x) |
( ) |
( ) |
( ) |
|
Is the research design appropriate? |
(x) |
( ) |
( ) |
( ) |
|
Are the methods adequately described? |
(x) |
( ) |
() |
( ) |
|
Are the results clearly presented? |
(x) |
( ) |
() |
( ) |
|
Are the conclusions supported by the results? |
(x) |
( ) |
( ) |
( ) |
Comments and Suggestions for Authors
The work "Pleurotus highking Mushroom Induces Apoptosis by Altering the Balance of Proapoptotic and Antiapoptotic Genes in Breast Cancer Cells and Inhibits Tumor Sphere Formation" presents very valuable research on anti-cancer activities of Pleurotus highking and perfectly fits into current research trends regarding the global problem of cancer. I recommend it for publication.
Response: Thank you very much. Your excellent comment will obviously appreciate us for future study. According to your suggestion, we checked and edited our manuscript by a professional manuscript writer.
Reviewer 2 Report
In my opinion searching for chemopreventive agents among mushroom species seems reasonable and necessary. The manuscript is well organised and interesting, however there is no information about the composition of the investigated fraction. Authors should add some determinations of chemical composition of this active fraction. Moreover, I suggest to change the title for shorter. Short name of the mushroom species should be P. highking.
Author Response
Reviewer 2
Open Review
(x) I would not like to sign my review report
( ) I would like to sign my review report
English language and style
( ) Extensive editing of English language and style required
( ) Moderate English changes required
( ) English language and style are fine/minor spell check required
(x) I don't feel qualified to judge about the English language and style
|
Yes |
Can be improved |
Must be improved |
Not applicable |
|
|
Does the introduction provide sufficient background and include all relevant references? |
(x) |
( ) |
( ) |
( ) |
|
Is the research design appropriate? |
() |
( ) |
(x) |
( ) |
|
Are the methods adequately described? |
(x) |
( ) |
() |
( ) |
|
Are the results clearly presented? |
(x) |
( ) |
() |
( ) |
|
Are the conclusions supported by the results? |
(x) |
( ) |
( ) |
( ) |
Comments and Suggestions for Authors
In my opinion searching for chemopreventive agents among mushroom species seems reasonable and necessary. The manuscript is well organised and interesting; however there is no information about the composition of the investigated fraction. Authors should add some determinations of chemical composition of this active fraction. Moreover, I suggest to change the title for shorter. Short name of the mushroom species should be P. highking.
Authors should add some determinations of chemical composition of this active fraction.
Response: Thank you very much for your excellent comments and constructive suggestions to improve the quality of our manuscript. We also think that it will be great if we can add some chemical composition of PEF-III. Now we are setting the experiments for phytochemical screening and did a collaboration to determine the structure of chemotherapeutic agent (s) from the reported fraction. We hope, we will be able to report its compositions with a complete structure in the near future. Thank you for your interest in it.
Moreover, I suggest to change the title for shorter. Short name of the mushroom species should be P. highking.
Response: We completely agree with you and included in our revised manuscript.
Reviewer 3 Report
This manuscript introduces to the evaluation of an active purified fraction from Pleurotus highking mushroom (PEF-III) as a potential anticancer agent in the human breast cancer cell line MCF-7, giving information to elucidate the possible mechanism of that activity. The experimental looks sound but there are some questions and corrections to be attended before considering accepting it for publication in Medicina.
Line 47 it should be “species”
Line 49 medicinal properties ? could be more specific?
Line 56-57 “Hence, we hypothesized that by supplying antioxidant and anti-inflammatory metabolites, P highking may reduce the risk of cancer.”
Section 2.3 Fractionation of extract
Considering reproducibility, authors should provide more details on the fractionation/purification operation of mushroom crude extract and from TLC analysis.
Why fraction number three (PEF-III) was selected from the five collected?
Were the other fractions also evaluated?
Did the crude extract was tested before?
Are there any suspected bioactive chemicals observed/reported in the mushroom to follow in the separation (give Rf values and spot colors).
How the PEF-III extract was prepared/suspended at the testing concentrations for the cell assays to assure no toxicity effect?
How the testing concentrations of PEF-III (20 and 30 μg/ml) were chosen ?
Line 77 it should be “… methanol in chloroform up to 100%”
Lines 79, 101, 158, 176, 199, 217, 233, 243, 251, Write numbers with letters
Lines 88, 97, 122, 130, 136, 164, 172, 190, 191, 196, 211, 230 Avoid repeating units/symbols e.g., “(20 and 30 μg/ml)”; “21 and 42%”
Line 104 “The morphological study was performed according to a published report [14].”
Line 144 Unit for minutes is “min”
Line 150-51 “In vitro tumor sphere assay was performed according to reported protocol [18].”
Author Response
Reviewer 3
Open Review
( ) I would not like to sign my review report
(x) I would like to sign my review report
English language and style
( ) Extensive editing of English language and style required
( ) Moderate English changes required
(x) English language and style are fine/minor spell check required
( ) I don't feel qualified to judge about the English language and style
|
Yes |
Can be improved |
Must be improved |
Not applicable |
|
|
Does the introduction provide sufficient background and include all relevant references? |
() |
(x) |
( ) |
( ) |
|
Is the research design appropriate? |
() |
( ) |
(x) |
( ) |
|
Are the methods adequately described? |
( ) |
(x) |
() |
( ) |
|
Are the results clearly presented? |
( ) |
(x) |
() |
( ) |
|
Are the conclusions supported by the results? |
() |
(x) |
( ) |
( ) |
Comments and Suggestions for Authors
This manuscript introduces to the evaluation of an active purified fraction from Pleurotus highking mushroom (PEF-III) as a potential anticancer agent in the human breast cancer cell line MCF-7, giving information to elucidate the possible mechanism of that activity. The experimental looks sound but there are some questions and corrections to be attended before considering accepting it for publication in Medicina.
Line 47 it should be “species”
Response: Thank you for your constructive comments to improve the quality of our manuscript. We wrote this line in a different way. Please see our revised manuscript.
Line 49 medicinal properties? Could be more specific?
Response: We revised this point like “They contain diverse biomolecules that are used for the treatment of various diseases including cancer”.
Line 56-57 “Hence, we hypothesized that by supplying antioxidant and anti-inflammatory metabolites, P highking may reduce the risk of cancer.”
Response: We included that in your revised manuscript.
Section 2.3 Fractionation of extract
Considering reproducibility, authors should provide more details on the fractionation/purification operation of mushroom crude extract and from TLC analysis.
Response: We included more information in this section.
Why fraction number three (PEF-III) was selected from the five collected?
Response: We select fraction number three (PEF-III) on the basis of its promising antiproliferative activity.
Were the other fractions also evaluated?
Response: We checked anticancer activity with the rest of the fractions and no significant results were observed with compared to control (DMSO).
Did the crude extract was tested before?
Response: Yes, we tested antibacterial, antioxidant, and brine shrimp lethality bioassay with crude extract of this mushroom and found significant results.
Are there any suspected bioactive chemicals observed/reported in the mushroom to follow in the separation (give Rf values and spot colors).
Response: To our best knowledge, no report is available about the separation and structural analysis of anticancer agents from our reported mushroom. Now we are setting the experiments for phytochemical screening and did a collaboration to determine the structure of chemotherapeutic agent (s) from the fraction. We hope, we will be able to report its compositions with a complete structure in the near future.
How the PEF-III extract was prepared/suspended at the testing concentrations for the cell assays to assure no toxicity effect?
Response: A stock solution of the fraction was prepared by dissolving in dimethyl sulfoxide (DMSO), which was further diluted with cell culture medium (DMEM). Then, test solutions were filtered through a 0.22 µm membrane filter and stored at -20oC for more experiments.
How the testing concentrations of PEF-III (20 and 30 μg/ml) were chosen?
Response: IC50 of PEF-III was 24 μg/ml. We selected slightly upper and lower concentration of IC50 to observe their effects on breast cancer cell lines.
Line 77 it should be “… methanol in chloroform up to 100%”
Response: Thank you; we corrected it in our revised manuscript.
Lines 79, 101, 158, 176, 199, 217, 233, 243, 251, Write numbers with letters
Response: Thank you; they are corrected in our revised manuscript.
Lines 88, 97, 122, 130, 136, 164, 172, 190, 191, 196, 211, 230 Avoid repeating units/symbols e.g., “(20 and 30 μg/ml)”; “21 and 42%”
Response: Thank you; they are corrected in our revised manuscript.
Line 104 “The morphological study was performed according to a published report [14].”
Response: Thank you; we revised it.
Line 144 Unit for minutes is “min”
Response: Yes; we revised it.
Line 150-51 “In vitro tumor sphere assay was performed according to reported protocol [18].”
Response: Thank you; we revised it.
Round 2
Reviewer 2 Report
I am glad that determination of chemical composition is in progress. I look forward for the next article.
In case of the title I was misunderstood. I did not specify clear enough. I meant that the whole title could (but not must) be shorter not the mushroom name in the title.
The suggestion that mushroom name should be P. highking concerned the Latin name of the mushroom used in the whole manuscript. The first letter of the genus name should be with dot and I think that it should be corrected.
Author Response
Reviewer 2
Open Review
(x) I would not like to sign my review report
( ) I would like to sign my review report
English language and style
( ) Extensive editing of English language and style required
( ) Moderate English changes required
( ) English language and style are fine/minor spell check required
(x) I don't feel qualified to judge about the English language and style
|
Yes |
Can be improved |
Must be improved |
Not applicable |
|
|
Does the introduction provide sufficient background and include all relevant references? |
(x) |
( ) |
( ) |
( ) |
|
Is the research design appropriate? |
() |
(X) |
() |
( ) |
|
Are the methods adequately described? |
(x) |
( ) |
() |
( ) |
|
Are the results clearly presented? |
(x) |
( ) |
() |
( ) |
|
Are the conclusions supported by the results? |
(x) |
( ) |
( ) |
( ) |
Comments and Suggestions for Authors
I am glad that determination of chemical composition is in progress. I look forward for the next article.
In case of the title I was misunderstood. I did not specify clear enough. I meant that the whole title could (but not must) be shorter not the mushroom name in the title. The suggestion that mushroom name should be P. highking concerned the Latin name of the mushroom used in the whole manuscript. The first letter of the genus name should be with dot and I think that it should be corrected.
In case of the title I was misunderstood. I did not specify clear enough. I meant that the whole title could (but not must) be shorter not the mushroom name in the title
Response: Thank you very much for your interest in our work. We are sorry for misunderstanding your point. In this stage, it is very difficult for us to shorten or change the manuscript title because it will require re-writing our conclusion according to title. Moreover, Medical English Communications Center, University of Tsukuba, Ibaraki, Japan thinks this title is appropriate according to our result and conclusion.
The suggestion that mushroom name should be P. highking concerned the Latin name of the mushroom used in the whole manuscript. The first letter of the genus name should be with dot and I think that it should be corrected.
Response: Thank you for your suggestion. We did it in our revised manuscript.
Reviewer 3 Report
Authors have made a good effort in attending observations from reviewers, therefore manuscript may be accepted for publication in Medicina. There is only a minor correction:
Line 18 (Abstract) you do not have to repeat the mushroom name
“… anticancer activity of Pleurotus highking mushroom purified extract …”
Author Response
Reviewer 3
Open Review
(X) I would not like to sign my review report
() I would like to sign my review report
English language and style
( ) Extensive editing of English language and style required
( ) Moderate English changes required
(x) English language and style are fine/minor spell check required
( ) I don't feel qualified to judge about the English language and style
|
Yes |
Can be improved |
Must be improved |
Not applicable |
|
|
Does the introduction provide sufficient background and include all relevant references? |
(X) |
() |
( ) |
( ) |
|
Is the research design appropriate? |
(X) |
( ) |
() |
( ) |
|
Are the methods adequately described? |
(X) |
() |
() |
( ) |
|
Are the results clearly presented? |
(X) |
() |
() |
( ) |
|
Are the conclusions supported by the results? |
(X) |
() |
( ) |
( ) |
Comments and Suggestions for Authors
Authors have made a good effort in attending observations from reviewers; therefore manuscript may be accepted for publication in Medicina. There is only a minor correction:
Line 18 (Abstract) you do not have to repeat the mushroom name“… anticancer activity of Pleurotus highking mushroom purified extract …”
Response: Thank you very much for your time and constructive suggestions to improve the quality of our manuscript. We revised this point in our manuscript.